# Importance of Sulfate Radical Anion Formation and Chemistry in Heterogeneous OH Oxidation of Sodium Methyl Sulfate, the Smallest Organosulfate

Kai Chung Kwong[1], Man Mei Chim[1], James F. Davies[2,4], Kevin R. Wilson[2], Man Nin Chan[1, 3]

[1]Earth System Science Programme, Faculty of Science, The Chinese University of Hong Kong, Hong Kong, CHINA
[2]Chemical Sciences Division, Lawrence Berkeley National Laboratory, Berkeley, USA
[3]The Institute of Environment, Energy and Sustainability, The Chinese University of Hong Kong, Hong Kong, CHINA
[4](now at) Department of Chemistry, UC Riverside, Riverside, USA

*Correspondence to*: Man Nin Chan (mnchan@cuhk.edu.hk)

**Abstract.** Organosulfates are important organosulfur compounds present in atmospheric particles. While the abundance, composition, and formation mechanisms of organosulfates have been extensively investigated, it remains unclear how they transform and evolve throughout their atmospheric lifetime. To acquire a fundamental understanding of how organosulfates chemically transform in the atmosphere, this work investigates the heterogeneous OH radical-initiated oxidation of sodium methyl sulfate ($CH_3SO_4Na$) droplets, the smallest organosulfate detected in atmospheric particles, using an aerosol flow tube reactor at a high relative humidity of 85 %. Aerosol mass spectra measured by a soft atmospheric pressure ionization source (Direct Analysis in Real Time, DART) coupled with a high-resolution mass spectrometer showed that neither functionalization nor fragmentation products are detected. Instead, the ion signal intensity of the bisulfate ion ($HSO_4^-$) has been found to increase significantly after OH oxidation. We postulate that sodium methyl sulfate tends to fragment into a formaldehyde ($CH_2O$) and a sulfate radical anion ($SO_4^{\bullet-}$) upon OH oxidation. The formaldehyde is likely partitioned back to the gas phase due to its high volatility. The sulfate radical anion, similar to OH radical, can abstract a hydrogen atom from neighboring sodium methyl sulfate to form the bisulfate ion, contributing to the secondary chemistry. Kinetic measurements show that the heterogeneous OH reaction rate constant, $k$, is $(3.79 \pm 0.19) \times 10^{-13}$ $cm^3$ $molecule^{-1}$ $s^{-1}$ with an effective OH uptake coefficient, $\gamma_{eff}$, of $0.17 \pm 0.03$. While about 40 % of sodium methyl sulfate is being oxidized at the maximum OH exposure ($1.27 \times 10^{12}$ molecule $cm^{-3}$ s), only a 3 % decrease in particle diameter is observed. This can be attributed to a small fraction of particle mass lost via the formation and volatilization of formaldehyde. Overall, we firstly demonstrate that the heterogeneous OH oxidation of an organosulfate can lead to the formation of sulfate radical anion and produce inorganic sulfate. Fragmentation processes and sulfate radical anion chemistry play a key role in determining the compositional evolution of sodium methyl sulfate during heterogeneous OH oxidation.

## 1 Introduction

Organosulfur compounds have been found to contribute a significant mass fraction of atmospheric organic compounds. A maximum organosulfur contribution of 30 % to $PM_{10}$ organic mass was estimated at a forest site in Hungary by calculating the difference between total sulfur and inorganic sulfate (*Surratt et al., 2008*). Using a similar approach, Tolocka and Turpin (2012) estimated that organosulfur compounds contribute up to 5 − 10 % of the total organic mass in southeastern United States, while Shakya and Peltier (2013) reported that organosulfur compounds account for about 1 − 2 % of organic carbon in Fairbanks, Alaska. Given their high atmospheric abundances, it is crucial to understand the composition, formation, and transformation of organosulfur compounds in the atmosphere.

Organosulfates have been identified as one of the major organosulfur compounds. The detection of organosulfates in laboratory studies have been complemented by a number of field observations, which confirm the presence of organosulfates in atmospheric particles (*Budisulistiorini et al., 2015; Chan et al., 2010; Darer et al., 2011; Frossard et al., 2011; Froyd et al., 2010; Hawkins et al., 2010; Hettiyadura et al., 2015; Huang et al., 2015; Iinuma et al., 2007; Kuang et al., 2016; Olson et al., 2011; Rattanavaraha et al., 2016; Riva et al., 2016; Shakya and Peltier, 2013; Stone et al., 2012; Surratt et al., 2007, 2008, 2010*). Various possible reaction pathways by which organosulfates form have been suggested. For instance, Iinuma et al. (2009) and Surratt et al. (2010) showed that organosulfates can be effectively formed via the reactive uptake of gas-phase epoxides, which are formed from the photooxidation of various biogenic volatile organic compounds (e.g. isoprene, α-pinene, and β-pinene), onto the acidic sulfate seed particles. Rudziński et al. (2009) and Nozière et al. (2010) suggested that the reactions between sulfate radical anion and reaction products of isoprene (e.g. methyl vinyl ketone and methacrolein) can yield a variety of organosulfates.

While the abundance, composition, and formation mechanisms have extensively been investigated, there is comparably little work understanding how organosulfates chemically transform in the atmosphere. Organosulfates are primarily present in the particle phase owing to their low volatility (*Huang et al., 2015; Estillore et al., 2016*). They can continuously react with gas-phase oxidants such as hydroxyl (OH) radicals, ozone ($O_3$), and nitrate ($NO_3$) radicals at or near the particle surface throughout their atmospheric lifetime. These heterogeneous oxidative processes have been found to change the size, composition, and physiochemical properties of both laboratory-generated organic particles and atmospheric particles (*Rudich et al., 2007; George and Abbatt, 2010; Kroll et al., 2015*). To gain a better understanding of how organosulfates chemically transform through heterogeneous oxidation in the atmosphere, this work investigates the heterogeneous OH radical-initiated oxidation of sodium methyl sulfate ($CH_3SO_4Na$) particles, the smallest organosulfate detected in atmospheric particles, using an aerosol flow tube reactor at a high relative humidity (RH) of 85 %. A soft atmospheric pressure ionization source (Direct Analysis in Real Time, DART) coupled with a high-resolution mass spectrometer was employed to characterize the molecular composition of the particles before and after OH oxidation in real time. Sodium methyl sulfate is detected in

atmospheric particles with a concentration of 0.7 ng m$^{-3}$ and 0.34 ng m$^{-3}$ during daytime and nighttime, respectively in Centreville, Alabama (*Hettiyadura et al., 2015*). As shown in **Table 1**, the simple structure of sodium methyl sulfate allows us to gain a more fundamental understanding of the heterogeneous oxidation kinetics and chemistry. The sodium salt of methyl sulfate could be considered as atmospherically relevant since a positive correlation between sodium ion and

organosulfates has been observed over the coastal areas (*Srooshian et al., 2015; Estillore et al., 2016*). The effects of salt (e.g. ammonium and potassium salt) on the heterogeneous oxidative kinetics and chemistry is also of atmospheric significance and warrants future study.

## 2 Experimental Method

An atmospheric pressure aerosol flow tube reactor was used to investigate the heterogeneous OH oxidation of sodium

methyl sulfate droplets. Detail experimental procedures have been described previously (*Davies and Wilson, 2015; Chim et al., 2017*). Briefly, aqueous droplets were generated by a constant output atomizer, and mixed with humidified nitrogen (N$_2$), oxygen (O$_2$), ozone (O$_3$), and hexane (a gas-phase OH tracer) before introducing into the reactor. The RH inside the reactor was maintained at 85 % and at a temperature of 20 ˚C. Estillore et al. (2016) measured the hygroscopicity of sodium methyl sulfate particles and showed that the particles absorb or desorb water reversibly upon increasing or decreasing RH. These

observations suggest that the particles likely exist as aqueous droplets over a range of RH (10 to 90 %). In our experiments, since the sodium methyl sulfate droplets are always exposed to a high RH, they are likely aqueous droplets prior to OH oxidation.

Sodium methyl sulfate droplets were oxidized inside the reactor by gas-phase OH radicals that were generated by the

photolysis of O$_3$ under ultraviolet light (254 nm) illumination in the presence of water vapor. The OH concentration was regulated by changing the O$_3$ concentration and determined by measuring the decay of hexane using gas chromatography coupled with a flame ionization detector. The OH exposure, defined as the product of OH concentration, [*OH*], and the particle residence time, *t*, was determined by measuring the decay of the gas-phase tracer, hexane (*Smith et al., 2009*).

$$OH\ Exposure = -\frac{\ln([Hex]/[Hex]_0)}{k_{Hex}} = \int_o^t [OH]dt \tag{1}$$

where [*Hex*] is the hexane concentration leaving the reactor, [*Hex*]$_0$ is the initial hexane concentration and $k_{Hex}$ is the second order rate constant of the gas-phase OH-hexane reaction. The aerosol residence time was determined to be 1.3 minutes, and the OH exposure was varied from 0 to 1.27 × 10$^{12}$ molecule cm$^{-3}$ s. The particle stream leaving the reactor was passed

through an annular Carulite catalyst denuder and an activated charcoal denuder to remove O$_3$ and gas-phase species, respectively.

A portion of the particle stream was sampled by a scanning mobility particle sizer (SMPS) for particle size distribution measurements. The remaining flow was delivered into a stainless-steel tube heater, where the particles were vaporized at 350 – 400 ˚C. Sodium methyl sulfate particles were confirmed to be fully vaporized upon heating at 300 ˚C or above by measuring the size distribution of the particles leaving the heater with the SMPS in a separate experiment. The resulting gas-

phase species were directed to an ionization region, a narrow open space between the DART ionization source (IonSense: DART SVP), and the inlet orifice of the high-resolution mass spectrometer (ThermoFisher, Q Exactive Orbitrap).

The details of the DART operation have been described elsewhere (*Cody et al., 2005*). The DART ionization source was operated in the negative-ion mode. Helium was chosen as the ionizing gas and entered an ionization chamber, where a high

electric potential of 4 kV was applied. This generates a glow discharge containing ions, electrons, and metastable helium atoms. A potential of 200 V was applied to two electrostatic lenses to remove ions and only the metastable helium atoms exited the chamber. The gas stream was heated to 500 ℃ before leaving the ionization source. The metastable helium atoms are responsible for ionizing the gas-phase species in the ionization region (*Chan et al., 2014; Cheng et al., 2015, 2016*). For ionic compounds like sodium methyl sulfate, negative ions can be formed via direct ionization in the negative ion mode

(*Hajslova et al., 2011*); for instance, pyruvate ions have been detected from ammonium pyruvate using the DART (*Block et al., 2010*).

We have run control experiments to investigate the potential volatilization of parent compound (i.e. sodium methyl sulfate), and the effect of ozone and UV light on the composition of the aerosols before oxidation under the same experimental

conditions. To investigate the volatilization of sodium methyl sulfate, we have measured the mass spectrum by filtering out the aerosols, and the parent peak is very small, suggesting there is very small amount of sodium methyl sulfate present in the gas phase. Volatilization and gas-phase oxidation of sodium methyl sulfate is expected to be not significant. For the effect of ozone and UV light, we found that there is no change in aerosol mass spectra in the presence of ozone without the UV light, suggesting that the reaction of sodium methyl sulfate with ozone is not significant. The aerosol mass spectrum is about the

same as that obtained in the absence of ozone with the UV light, suggesting that the photolysis of sodium methyl sulfate aerosols is not likely to occur.

### 3 Results and Discussions

### 3.1 Aerosol Mass Spectra

**Figure 1** shows the aerosol mass spectra before and after oxidation. Before oxidation (**Figure 1a**), there is one major peak

and some minor background peaks. The largest peak at $m/z$ 111 has a chemical formula of $CH_3SO_4^-$, which is corresponding to the negative ion (i.e. anionic form) of sodium methyl sulfate. For ionic compounds, negative ions can be formed via direct ionization in the negative ion mode (*Hajslova et al.,2011*). For instance, pyruvate ions have been detected from the

ammonium pyruvate using the DART (*Block et al., 2010*). **Figure 1b** shows that the intensity of the parent compound decreases after oxidation. At the maximum OH exposure ($1.27 \times 10^{12}$ molecule cm$^{-3}$ s), only one new peak at *m/z* 97 evolves, corresponding to the bisulfate ion (HSO$_4^-$). As shown in **Figure 2**, its intensity increases significantly after oxidation, suggesting that the bisulfate ion is likely generated during the oxidation. Based on the aerosol speciation data measured at different extents of OH oxidation, oxidation kinetics will be determined in section 3.2 and reaction mechanisms will be proposed in section 3.3 to explain the formation of major ions detected in the aerosol mass spectra.

### 3.2 Oxidation Kinetics

The normalized parent decay as a function of OH exposure is shown in **Figure 3** and the OH radical-initiated decay can be fitted using an exponential function:

$$ln\frac{I}{I_0} = -k\,[OH]\cdot t \tag{2}$$

where $I$ is the ion signal at a given OH exposure, $I_0$ is the ion signal before oxidation, $k$ is the second order heterogeneous rate constant, and $[OH] \cdot t$ is the OH exposure. The exponential $k$ is determined to be $(3.79 \pm 0.19) \times 10^{-13}$ cm$^3$ molecule$^{-1}$ s$^{-1}$. Assuming a 24-hour average OH concentration of $1.5 \times 10^6$ cm$^{-3}$, the lifetime of sodium methyl sulfate against heterogeneous OH oxidation is about 20 days. This timescale is longer than other removal processes such as wet or dry deposition. Laboratory studies have revealed that primary and secondary organosulfates are stable against hydrolysis under atmospheric relevant aerosol acidities and lifetimes, while tertiary organosulfates may undergo hydrolysis efficiently (*Hu et al., 2011*). Since sodium methyl sulfate is a primary organosulfate, it is expected to be stable against hydrolysis. These results suggest that sodium methyl sulfate is likely chemically stable over atmospherically relevant timescales. Studying the heterogeneous reactivity of sodium methyl sulfate towards OH radicals provides a much needed fundamental understanding of the oxidation kinetics and pathways, and these data may be applied in the interpretation of the oxidation of more complex organosulfates, which may have a range of chemical lifetimes in the atmosphere. The effective uptake coefficient, $\gamma_{eff}$, defined as the fraction of OH collisions that yield a reaction, is computed (*Davies and Wilson, 2015*),

$$\gamma_{eff} = \frac{2}{3}\frac{D_0\,\rho\,mfs\,N_A}{M_w\,\overline{c_{OH}}}\,k \tag{3}$$

where $D_0$ is the mean surface-weighted particle diameter, $\rho$ is the aerosol density before oxidation, *mfs* is the mass fraction of solute, $N_A$ is the Avogadro's number, $M_w$ is the molecular weight of sodium methyl sulfate, and $\overline{c_{OH}}$ is the average speed of gas-phase OH radicals. The mean surface-weighted particle diameter was 218 nm and decreased slightly to 211 nm (about 3 % decrease) at the maximum OH exposure (**Figure 4**). Before oxidation, the composition of the droplets (i.e. *mfs*) is derived from the hygroscopicity data reported by Estillore et al. (2016). The particle growth factor, $G_f$, defined as the ratio of the diameter at different RH to the dry particle at a reference RH (RH$_1$), is converted into *mfs* using the following equation (*Ansari and Pandis, 2000; Peng et al., 2001*),

$$G_f = \left(\frac{mfs_{RH,1}\,\rho_{RH,1}}{mfs_{RH,2}\,\rho_{RH,2}}\right)^{\frac{1}{3}} \tag{4}$$

where $mfs_{RH,i}$ and $\rho_{RH,i}$ are the mass fraction of solute and particle density at a given RH, respectively. It is assumed that sodium methyl sulfate exists as an anhydrous particle at the reference RH (RH < 10 %) (i.e. $mfs_{RH,1} = 1$). The particle density is estimated using the volume additivity rule with the density of water and sodium methyl sulfate (1.60 g cm$^{-3}$, *Chemistry Dashboard*) with an uncertainty of 20 – 30 %. The *mfs* is computed to be 0.34 at 85 %. Using **Eqn. 3**, the $\gamma_{eff}$ is calculated to be 0.17 ± 0.03. Although the $\gamma_{eff}$ is less than 1, as will be discussed in the section 3.3, secondary reactions are likely occurring, leading to the formation and subsequent reactions of sulfate radical anions ($SO_4^{\bullet-}$).

### 3.3 Reaction Mechanisms: OH Reaction with Sodium Methyl Sulfate

Sodium methyl sulfate tends to dissociate and exist in its ionic form because of its high dissociation constant (pKa = −2.4). As shown in **Scheme 1**, the oxidation is initiated by hydrogen abstraction from the methyl group by the OH radical, forming an alkyl radical that quickly reacts with an oxygen molecule to form a peroxy radical. Based on the well-known particle-phase reactions (*George and Abbatt, 2010*), the self-reaction of two peroxy radicals can form a carbonyl functionalization product ($CHSO_5^-$) via the Bennett and Summers mechanism (*Bennett and Summers, 1974*), or form both alcohol ($CH_3SO_5^-$) and carbonyl functionalization products via the Russell reactions (*Russell, 1957*). Alternatively, alkoxy radicals can be produced through the peroxy–peroxy radical reactions. Once formed, alkoxy radical can react with an oxygen molecule to form the carbonyl functionalization product or abstract a hydrogen atom from the neighboring molecules to form the alcohol functionalization product. Furthermore, the alkoxy radical can undergo fragmentation to yield a formaldehyde ($CH_2O$) and a sulfate radical anion ($SO_4^{\bullet-}$). Alternatively, the decomposition of the alkoxy radical, in analogy to the OH radical-initiated oxidation of simple alkyl esters (*Sun et al., 2012*), could involve the rearrangement of the hydrogen atom from the alkoxy radical carbon to the oxygen on the methoxy group. The decomposition of the C−O bond from the methoxy group generates a bisulfate ion and a formyl radical (CHO). The subsequent reactions of the formyl radical can yield a carbon monoxide (CO) and $HO_2$. Like formaldehyde, carbon monoxide is volatile and partitions back to the gas phase.

As shown in **Figure 1**, neither functionalization nor fragmentation products are detected. Formaldehyde has a mass which is below the mass range of the mass spectrometer and, once formed, it is likely partitioned back to the gas phase due to its high volatility. On the other hand, it is expected that the two functionalization products can be detected by DART ionization source if they were formed in significant amount. Additional experiments were performed to verify whether the alcohol and carbonyl functionalization products can be detected by the DART ionization source. We have measured the heterogeneous OH radical-initiated oxidation of sodium ethyl sulfate ($C_2H_5SO_4Na$) under similar experimental conditions. As shown in **Figure 5**, the negative ions of the alcohol ($C_2H_5SO_5^-$) and carbonyl ($C_2H_3SO_5^-$) functionalization products are detected in the aerosol mass spectra. When the sodium ethyl sulfate is oxidized (**Figure 6**), the abundance of these two functionalization products increases with increasing OH exposure (**Figure 7**). Similar to sodium methyl sulfate, the bisulfate ion has been detected and its intensity increases after oxidation (**Figure 8**). These results suggest that if functionalization products are formed during the OH oxidation of sodium methyl sulfate, they could be detected by DART ionization source.

The absence of the functionalization products in the aerosol mass spectra suggests that the OH reaction with sodium methyl sulfate tends to undergo fragmentation processes rather than functionalization processes. One possibility is that, due to the presence of bulky sulfate group relative to the methyl group, reaction intermediates resulted from the self-reaction of two peroxy radicals may not be easily arranged into appropriate configurations (i.e. cyclic transition states), which are required for the formation of the functionalization products via Russell reaction or Bennett and Summers mechanism. Alternatively, alkoxy radicals are more likely formed, followed by fragmentation. Although the dissociation energies for C−O and C−C bonds of sodium methyl sulfate are not known, fragmentation processes could be enhanced since the decomposition of the alkoxy radical involves the cleavage of a C−O bond, which is in general thought to be weaker than a C−C bond (*Dean and Lang, 1992*). The bond dissociation energy of a C−O bond might be lowered in the presence of a sulfur atom or sulfur-oxygen bearing group (*Oae and Doi, 1991; Dean and Lange, 1992*). One possibility is that sulfur atom or sulfur-oxygen bearing group (e.g. sulfate) is more electronegative than a carbon atom, reducing the electron density and bond strength of the C−O bond by inductive effect. Further investigation is required to better understand the effect of sulfate group on the dissociation energies of the C−O bond for the sodium methyl sulfate.

**3.4 Formation and Reaction of Sulfate Radical Anion in the OH Reaction with Sodium Methyl Sulfate**

**Scheme 1** shows that sulfate radical anion ($SO_4^{\bullet-}$) can be formed via the fragmentation processes. The sulfate radical anion is a strong oxidant in aqueous phase. For example, Huie and Clifton (1989) have reported that hydrogen abstraction by sulfate radical anions on the alkane can result in the formation of bisulfate ions. They also reported that the hydrogen abstraction rate is the highest on the tertiary carbon, and the rate is one order of magnitude smaller for the secondary carbon, and even smaller for the primary carbon. The second-order rate constants for $SO_4^{\bullet-}$ reactions with alcohols, ethers, alkanes, and aromatic compounds typically range in value from $10^6$ to $10^9$ $M^{-1}$ $s^{-1}$ (*Clifton and Huie, 1989; Neta et al., 1977; Neta et al., 1988; Padmaja et al., 1993*). With an aqueous phase $SO_4^{\bullet-}$ concentration of $10^{-14}$ M (*Herrmann et al., 2000*), the calculated lifetime toward aqueous phase oxidation with $SO_4^{\bullet-}$ ranges from 1.2 days to 3 years. These results suggest that some organic compounds (e.g. alkanes and alkenes) are stable against $SO_4^{\bullet-}$ initiated reactions, but some (e.g. alcohols and ethers) can react with $SO_4^{\bullet-}$ efficiently. Future works are needed to better understand the role of $SO_4^{\bullet-}$ initiated oxidation chemistry in chemical transformation of sodium methyl sulfate and organic compounds in the atmospheric aerosols. For the OH reaction with sodium methyl sulfate, it is proposed that sulfate radical anion, once formed, can abstract a hydrogen atom from the neighboring, unreacted sodium methyl sulfate, yielding the bisulfate ion, which has a small acid dissociation constant in equilibrium with sulfate ($SO_4^{2-}$) and hydrogen ($H^+$) ions ($Ka = 1.2 \times 10^{-2}$) (*Brown et al., 2012*).

$$SO_4^{\bullet-} + CH_3SO_4^- \rightarrow CH_2SO_4^{\bullet-} + HSO_4^- \tag{5}$$

$$HSO_4^- \rightleftharpoons H^+ + SO_4^{2-} \tag{6}$$

Moreover, the sulfate radical anion may react with particle-phase water to form a bisulfate ion and an OH radical (*Tang et al., 1988*).

$$SO_4^{\bullet-} + H_2O \rightleftharpoons OH^{\bullet} + HSO_4^- \tag{7}$$

As shown in **Figure 1b**, the bisulfate ion is the second largest peak detected in the aerosol mass spectrum and its intensity has found to increase significantly with increasing OH exposure (**Figure 2**). The detection of the bisulfate ion provides indirect evidence to support the formation and subsequent reactions of sulfate radical ions. When these reactions occur (**Eqn. 5 and 7**), additional sodium methyl sulfate is consumed by the sulfate radical ions and OH radicals, contributing to the secondary chemistry. It is also known that the self-reaction of two sulfate radical anions can yield a peroxydisulfate ion ($S_2O_8^{2-}$) (*Hayon et al., 1972; Tang et al., 1988; Huie et al., 1989; Huie et al., 1991; Das, 2001*):

$$SO_4^{\bullet-} + SO_4^{\bullet-} \rightarrow S_2O_8^{2-} \tag{8}$$

Based on its mass-to-charge ratio, the peroxydisulfate ion can be detected as $SO_4^-$ at *m/z* 96 in the aerosol mass spectra (**Figure 1**). It is worth noting that the peak at *m/z* 96 does not likely originate from the sulfate radical anions due to its high reactivity. The ion signal intensity of the $SO_4^-$ is measured to be smaller than that of the bisulfate ion (**Figure 2**). However, the abundance of these two ions cannot be directly inferred from their intensities owing to their unknown ionization efficiencies in the DART ionization source.

### 3.5 Sodium Methyl Sulfate vs. Sodium Ethyl Sulfate: Kinetics and Chemistry

We here further examine the results of sodium methyl sulfate and sodium ethyl sulfate to gain more insights into how the carbon number affects the kinetics and chemistry for these two small organosulfates ($C_1$ and $C_2$). Kinetic measurements show that the heterogeneous rate constant and effective OH uptake coefficient of sodium ethyl sulfate are determined to be $(4.64 \pm 0.29) \times 10^{-13}$ cm$^3$ molecule$^{-1}$ s$^{-1}$ and $0.19 \pm 0.03$, respectively (**Table 1** and **Figure 6**). These kinetic parameters are slightly larger than that of sodium methyl sulfate ($3.79 \pm 0.19 \times 10^{-13}$ cm$^3$ molecule$^{-1}$ s$^{-1}$ and $0.17 \pm 0.03$). An additional carbon atom does not significantly change the heterogeneous OH reactivity. On the other hand, the composition of the sodium ethyl sulfate (**Figure 5**) is different from that of sodium methyl sulfate after oxidation (**Figure 1**). As discussed in section 3.3, the bisulfate ion ($HSO_4^-$) and $SO_4^-$ have been observed for both organosulfates. However, the alcohol and carbonyl functionalization products are only detected in the OH oxidation of sodium ethyl sulfate. These observations suggest the potential reaction pathways may change with an increasing carbon number.

As shown in **Scheme 2**, at the first OH oxidation step of sodium ethyl sulfate, the hydrogen abstraction can occur either on the primary (**Scheme 2, Path A**) or the secondary carbon site (**Scheme 2, Path B**). Depending on the initial OH reaction site, two structural isomers of alcohol ($C_2H_5SO_5^-$) and carbonyl ($C_2H_3SO_5^-$) functionalization products can be formed. However, these isomers cannot be differentiated by exact mass measurements. Although the preferential OH reaction site is not well understood, we postulate that the formation of the alcohol and carbonyl functionalization products are likely originated from the hydrogen abstraction occurred at the primary carbon (**Scheme 2, Path A**). One likely explanation is that based on the knowledge of the OH reaction with sodium methyl sulfate (**Scheme 1**), when the hydrogen abstraction occurs at a carbon atom adjacent to the sulfate group, an alkoxy radical is likely formed from the self-reaction of two peroxy radicals and tends

to decompose. It is hypothesized that when the hydrogen atom of the secondary carbon is abstracted by the OH radical (**Scheme 2, Path B**), similar to the sodium methyl sulfate, an alkoxy radical is likely generated and fragments into a sulfate radical anion and an acetaldehyde, which is volatile and likely partitions back to the gas phase. The sulfate radical anion can subsequently react with an unreacted sodium ethyl sulfate, leading to the formation of a bisulfate ion (HSO$_4^-$, *m/z* 97).

Alternatively, the self-reactions of two sulfate radical anions can yield a peroxydisulfate ion (S$_2$O$_8^{2-}$), which can be detected as SO$_4^-$ at *m/z* 96 in the aerosol mass spectra. Future works are needed to verify these hypotheses.

### 3.6 Aerosol Mass Lost via Volatilization

While the fragmentation and volatilization processes are likely the dominant reaction pathways of OH oxidation of sodium methyl sulfate, the diameter of the particles decreases slightly from 218 nm to 211 nm at the maximum OH exposure

(**Figure 4**). As shown in **Scheme 1,** when the fragmentation processes occur, one methyl group is lost via volatilization in the form of formaldehyde. The methyl group (CH$_3$, $M_w$ = 15 g mol$^{-1}$) contributes about 11 % of the total molecular mass (CH$_3$SO$_4$Na, $M_w$ = 134 g mol$^{-1}$). At the maximum OH exposure, about 40 % of sodium methyl sulfate is reacted (**Figure 3**). If we assume that only fragmentation processes occur during OH oxidation, this will lead to a 4.4 % loss in particle mass via volatilization. The result of this simple analysis is consistent with the experimental observation that only a small decrease in

particle size (~ 3 %) is measured after oxidation. For the sodium ethyl sulfate (**Figure 9**), the particle diameter decreases slightly from 203 nm to 195.5 nm at the maximum OH exposure (~ 4 % decrease in particle diameter). According to **Scheme 2**, formaldehyde and acetaldehyde are the volatile fragmentation products, which are likely partitioned back to the gas phase. Following the above analysis, if we assume the fragmentation only leads to the formation and volatilization of the acetaldehyde, this will lead to a maximum 9 % loss in the particle mass at the highest OH exposure. Similar to sodium

methyl sulfate, the formation and volatilization of fragmentation products do not cause a significant decrease in particle mass (and diameter) during OH oxidation.

### 4 Conclusions and Atmospheric Implications

This work investigates the heterogeneous OH oxidation of sodium methyl sulfate, the smallest organosulfate found in atmospheric particles. During oxidation, sodium methyl sulfate tends to fragment into a formaldehyde and a sulfate radical

anion. The formation and chemistry of sulfate radical anions in the heterogeneous OH oxidation of organosulfates could be of atmospheric interest. This is because sulfate radical anion, like OH radical, can abstract a hydrogen atom from unreacted sodium methyl sulfate, contributing to the secondary reactions. The formation of bisulfate and likely sulfate ions from sulfate radical anion reactions suggest that OH reaction with sodium methyl sulfate or other organosulfates can possibly lead to the formation of inorganic sulfate. Moreover, sulfate radical anions can react with organic compounds to regenerate

organosulfates. Compared to sodium methyl sulfate, the OH reaction with sodium ethyl sulfate occurs at a similar reaction rate. The oxidation of both compounds can lead to the formation of bisulfate ions, but different distribution of reaction

products is observed. These observations suggest that the carbon number plays a significant role in governing the reaction mechanisms for these two small organosulfates. Given a variety of organosulfates have been detected in atmospheric particles, the role of molecular structure (e.g. carbon chain length, position and nature of functional groups (e.g. hydroxyl and carbonyl)) in the heterogeneous OH oxidation kinetics, chemistry, and sulfate radical anion formation and reactions of

organosulfates remains unexplored and warrants further study.

## 5 Acknowledgement

K. C. Kwong, M. M. Chim, and M. N. Chan are supported by the Direct Grant for Research (4053159), The Chinese University of Hong Kong and Hong Kong Research Grants Council (HKRGC) Project ID: 2191111 (Ref 24300516). J. F. Davies and K. R. Wilson are supported by the Director, Office of Energy Research, Office of Basic Energy Sciences,

Chemical Sciences, Geosciences, and Biosciences Division, in Condensed Phase and Interfacial Molecular Science Program of the U.S. Department of Energy under Contract No. DE-AC02-05CH11231.

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

**Table 1. Chemical structures, properties, rate constant, and effective OH uptake coefficient of sodium methyl sulfate and sodium ethyl sulfate**

| Compounds | Sodium Methyl Sulfate | Sodium Ethyl Sulfate |
|---|---|---|
| Chemical Formula |  |  |
| Molecular Formula | $CH_3SO_4Na$ | $C_2H_5SO_4Na$ |
| Molecular Weight | 134.0867 | 148.1147 |
| Density $(g\ cm^{-3})$ | 1.60 | 1.46 |
| Mass Fraction of Solute, $mfs$ at 85 % RH | 0.34 | 0.38 |
| Heterogeneous OH Rate Constant, $k$ $(\times 10^{-13}\ cm^3\ molecule^{-1}\ s^{-1})$ | $3.79 \pm 0.19$ | $4.64 \pm 0.29$ |
| Effective OH Uptake Coefficient, $\gamma_{eff}$ | $0.17 \pm 0.03$ | $0.19 \pm 0.03$ |

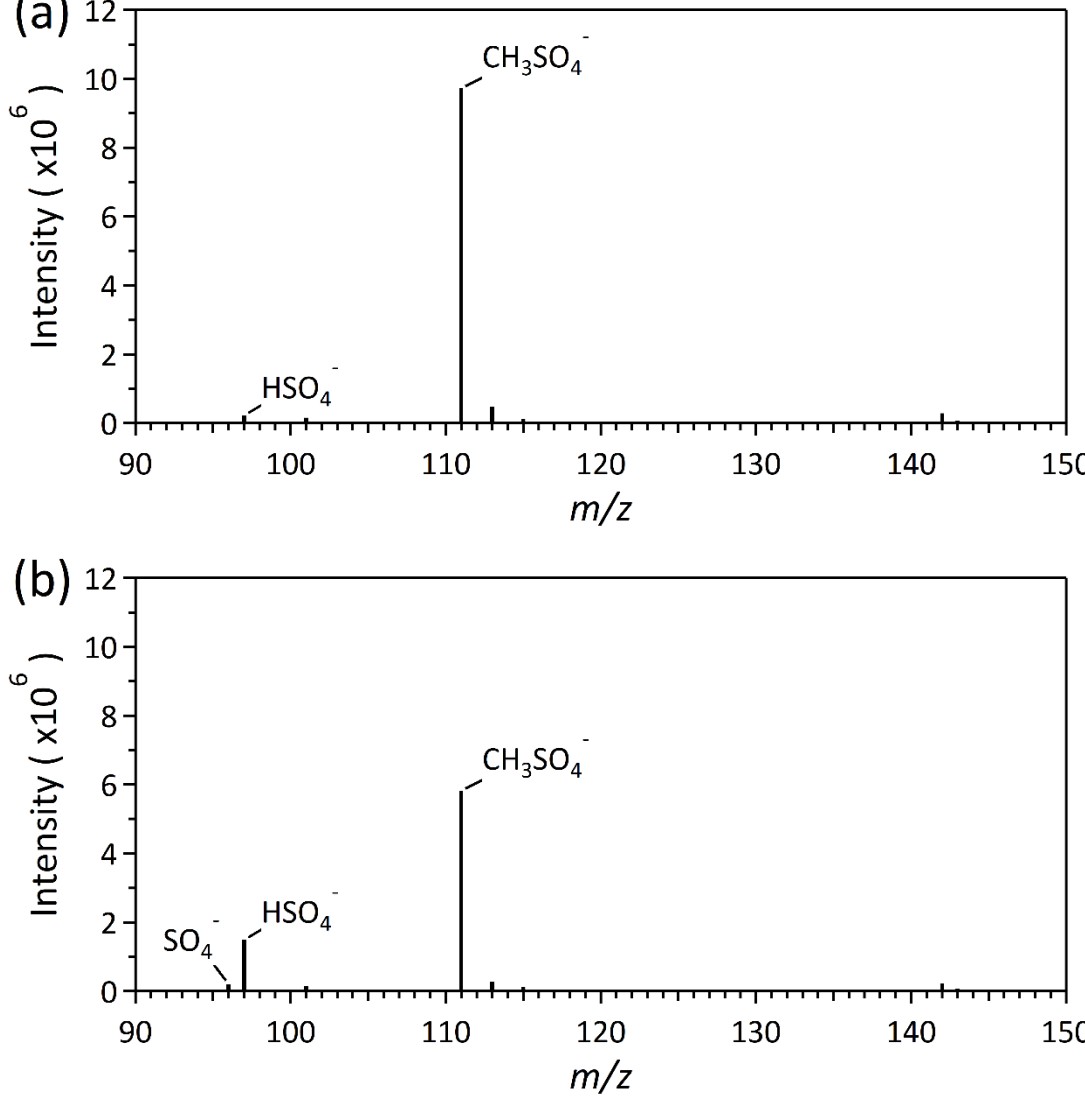

**Figure 1. Aerosol mass spectra of sodium methyl sulfate before (a) and after (b) oxidation**

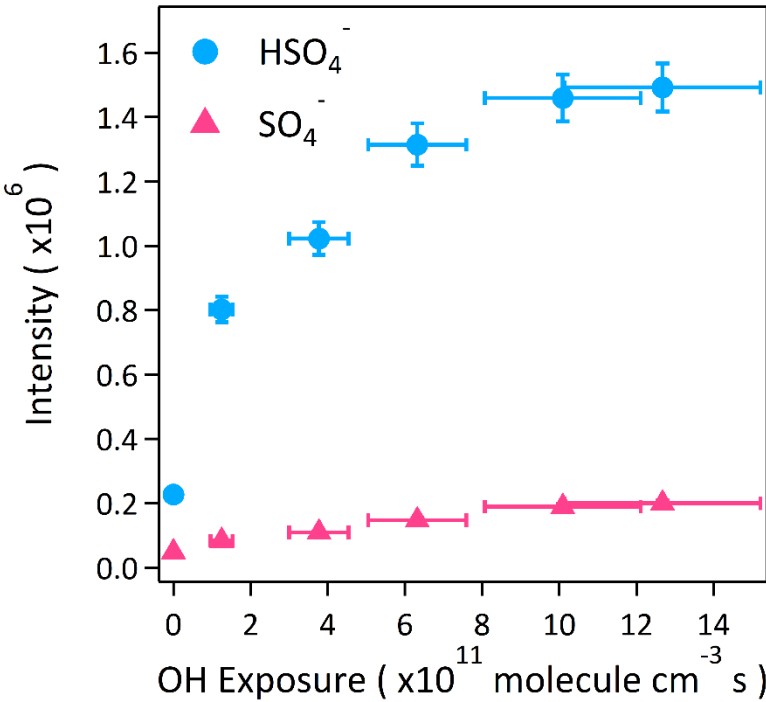

**Figure 2. The kinetic evolution of $HSO_4^-$ and $SO_4^-$ as a function of OH exposure during the heterogeneous OH oxidation of sodium methyl sulfate. The small uncertainty in ion intensity measurement for $SO_4^-$ is not visualized in the figure.**

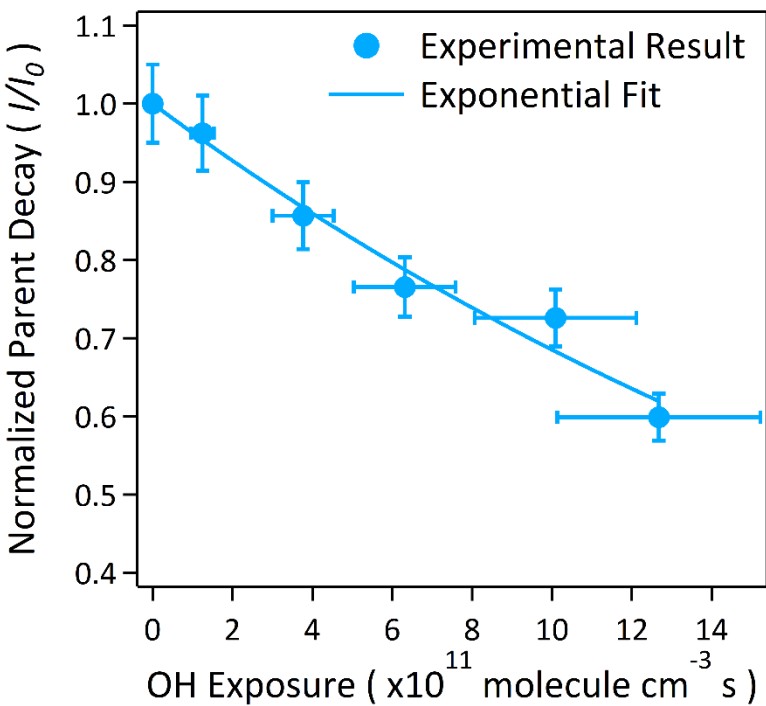

**Figure 3. The normalized decay of sodium methyl sulfate as a function of OH exposure during the heterogeneous OH oxidation**

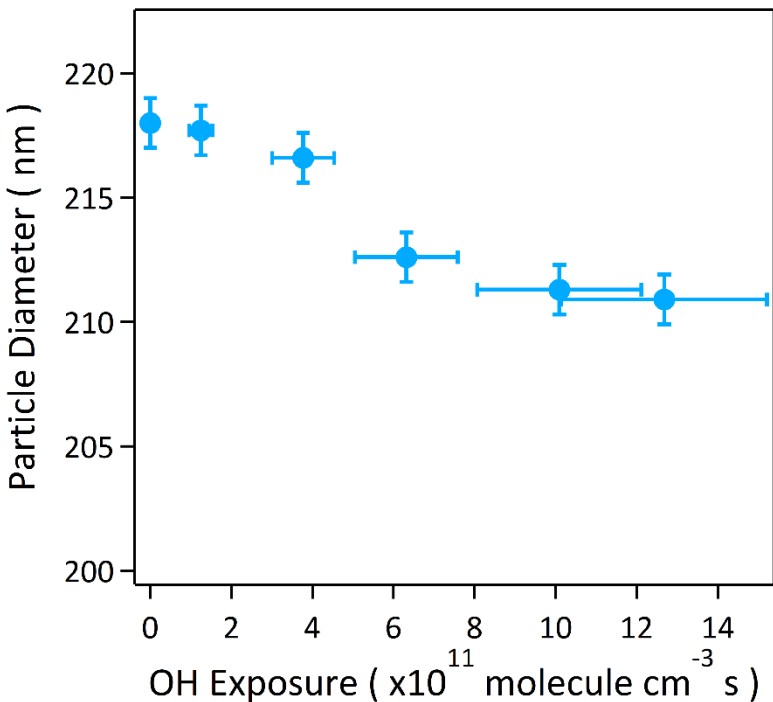

**Figure 4. The surface-weighted particle diameter of sodium methyl sulfate as a function of OH exposure during heterogeneous OH oxidation**

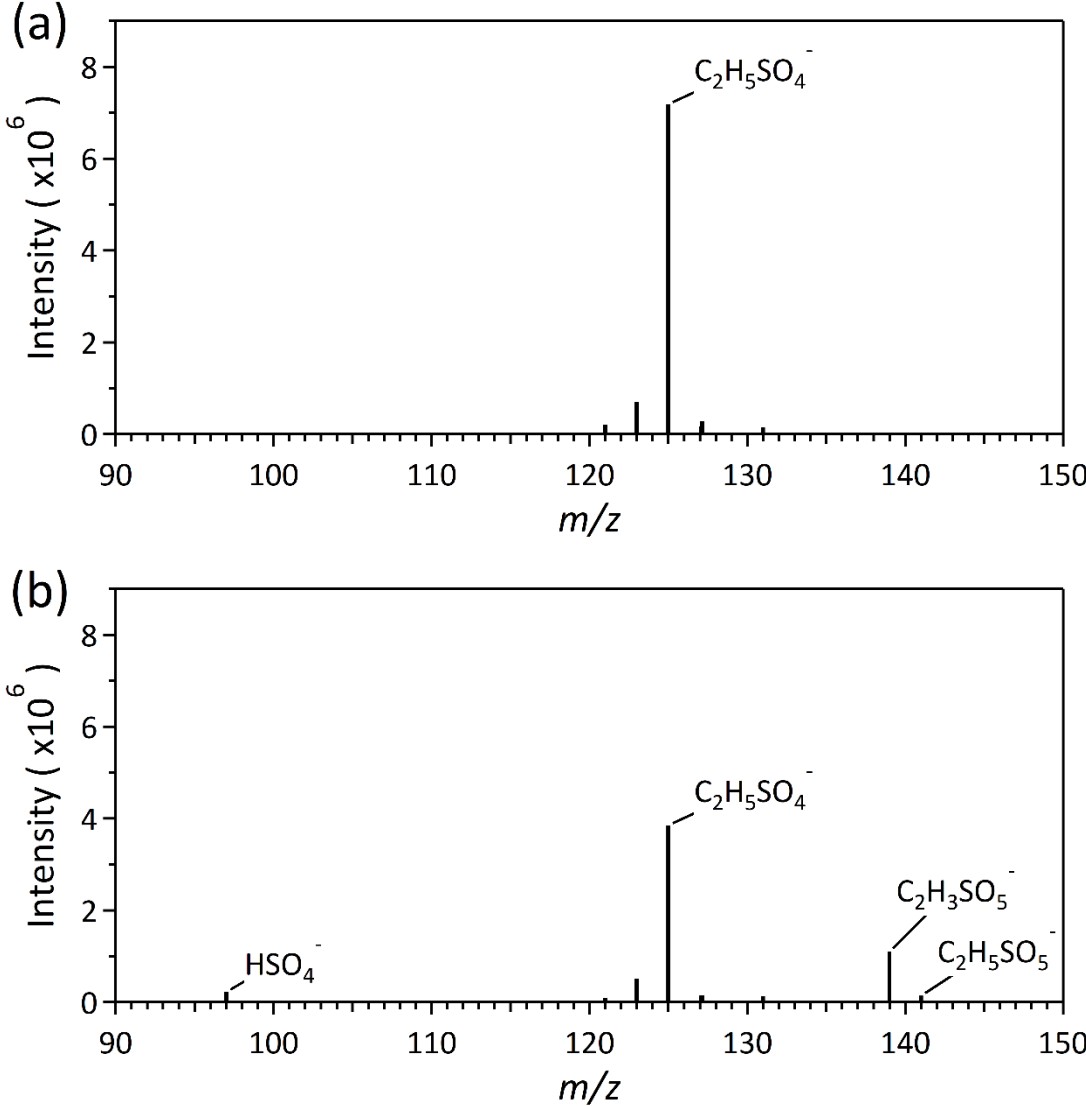

**Figure 5. Aerosol mass spectra of sodium ethyl sulfate before (a) and after (b) oxidation**

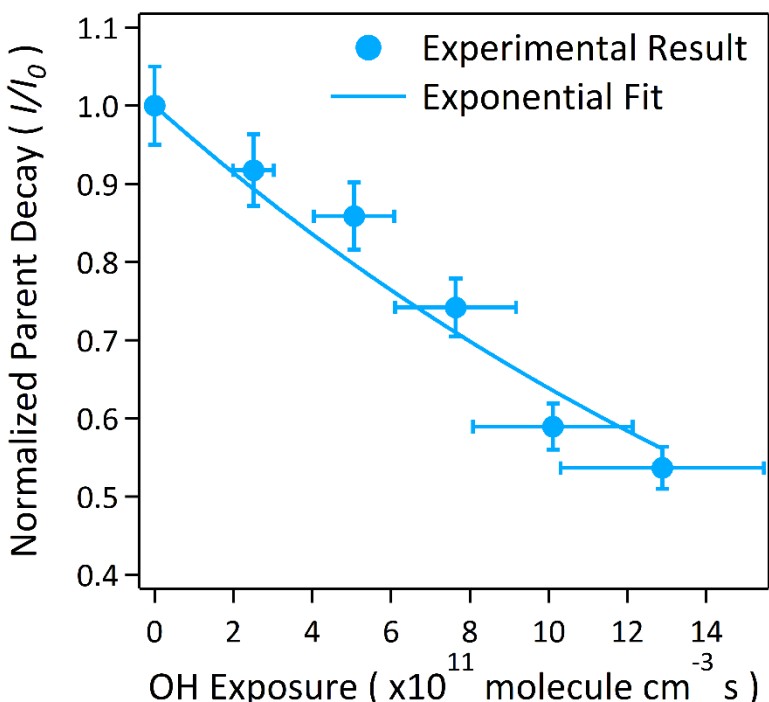

**Figure 6. The normalized parent decay of sodium ethyl sulfate as a function of OH exposure in the heterogeneous OH oxidation**

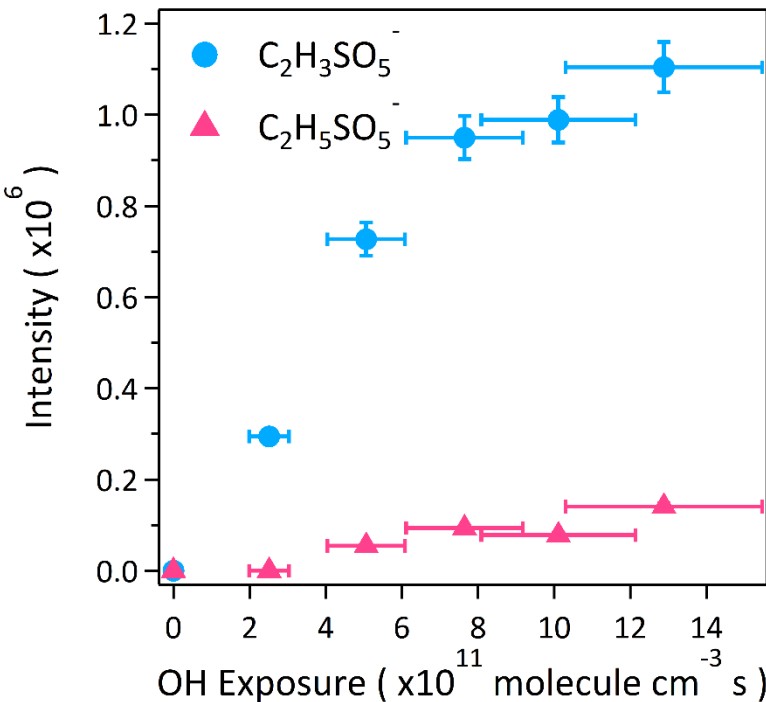

**Figure 7. The kinetic evolution of carbonyl ($C_2H_3SO_5^-$) and alcohol ($C_2H_5SO_5^-$) functionalization products as a function of OH exposure in the heterogeneous OH oxidation of sodium ethyl sulfate. The small uncertainty in ion intensity measurement for $C_2H_5SO_5^-$ is not visualized in the figure.**

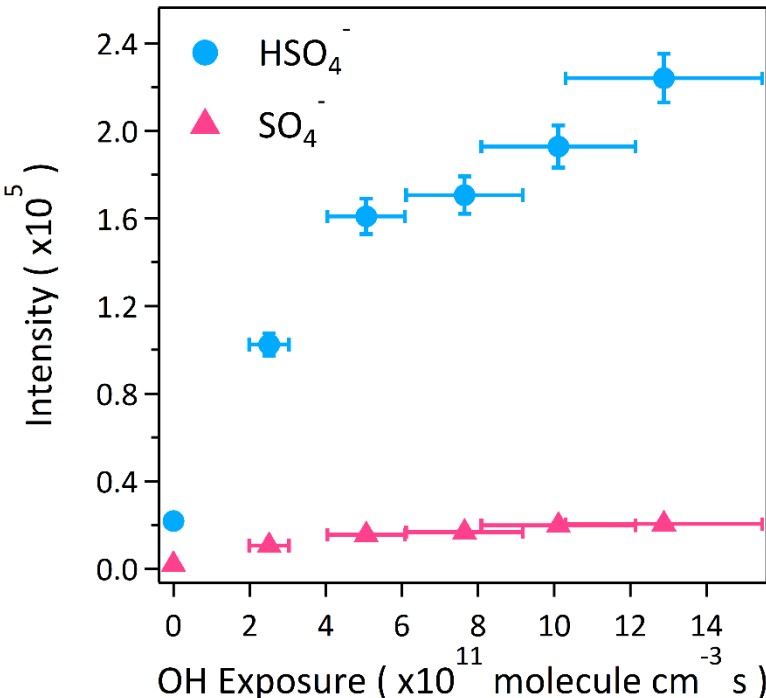

**Figure 8. The kinetic evolution of $HSO_4^-$ and $SO_4^-$ as a function of OH exposure in the heterogeneous OH oxidation of sodium ethyl sulfate. The small uncertainty in ion intensity measurement for $SO_4^-$ is not visualized in the figure.**

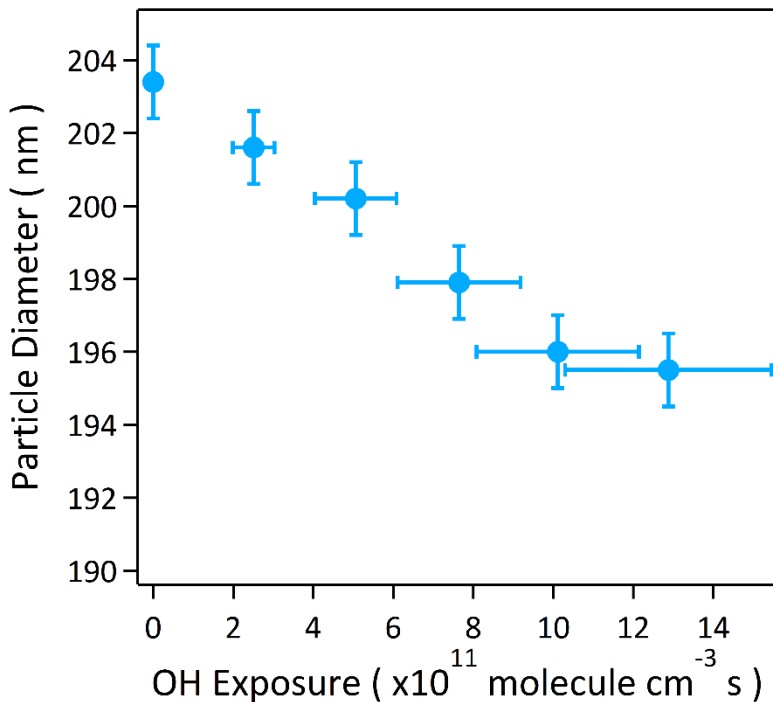

**Figure 9. The surface-weighted particle diameter of sodium ethyl sulfate as a function of OH exposure during heterogeneous OH oxidation**

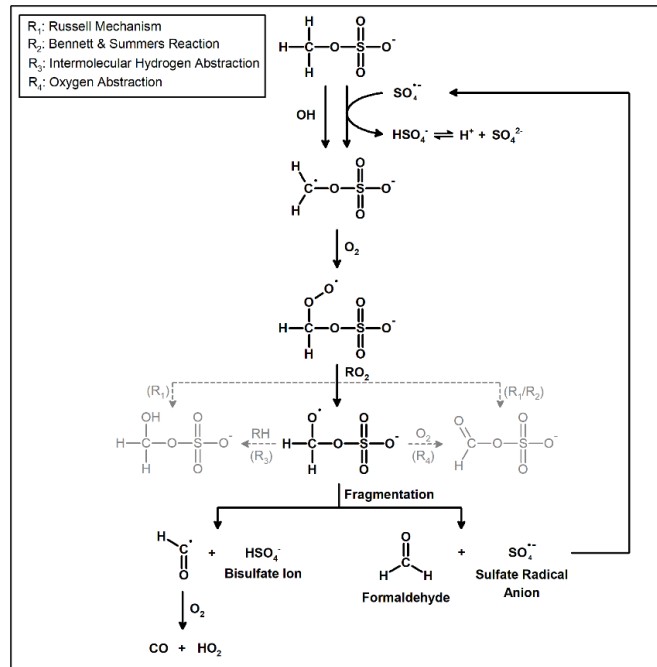

**Scheme 1. Proposed reaction mechanism for heterogeneous OH oxidation of sodium methyl sulfate (Gray arrows denote the minor pathways)**

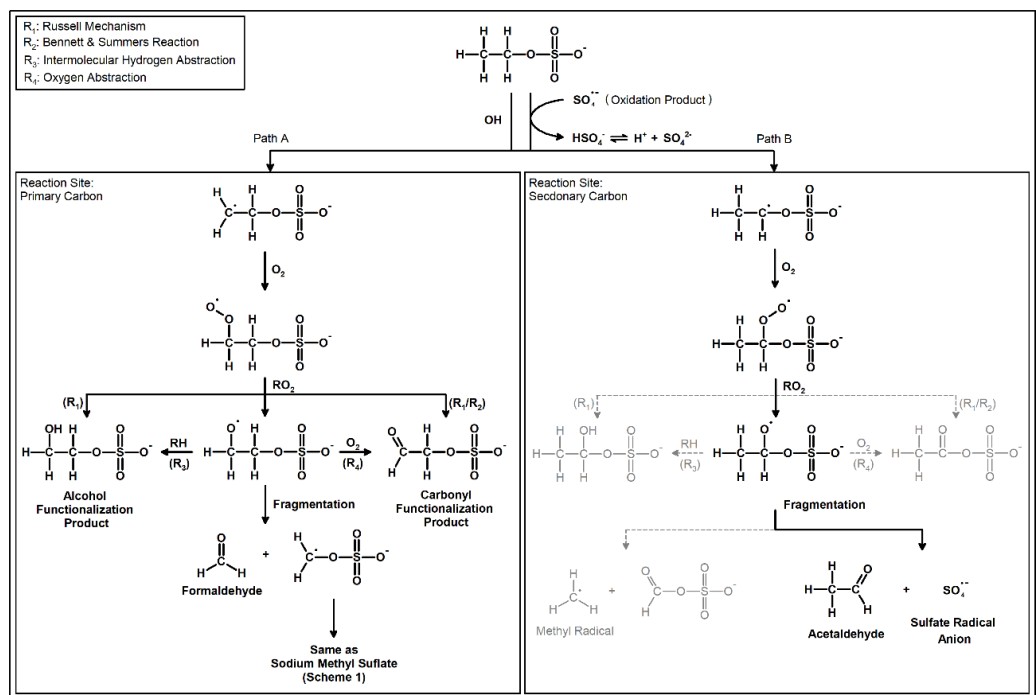

5    **Scheme 2. Proposed reaction mechanism of heterogeneous OH oxidation of sodium ethyl sulfate (Gray arrows denote the minor pathways)**