# Peer review of "Importance of Sulfate Radical Anion Formation and Chemistry in Heterogeneous OH Oxidation of Sodium Methyl Sulfate, the Smallest Organosulfate"

_Atmospheric Chemistry and Physics, 2017_

## Referee Comment (RC1) · Anonymous Referee #3 · 20 Nov 2017

**Overview**

In this paper, titled "Importance of sulfate radical anion formation and chemistry in heterogeneous OH oxidation of sodium methyl sulfate, the smallest organosulfate" by Kwong et al., the authors present an interesting dataset focused on the heterogeneous chemical transformation of an organic sulfate compound. Organosulfates often have low saturation vapor pressure and have been identified in ambient aerosol particles, and therefore are considered important for SOA formation in the atmosphere. However, there is currently very little knowledge of the particle phase transformations of organosulfates, which could be important because variations in the organosulfate composition may influence particle volatility and SOA mass. Therefore, the topic is very much atmospherically relevant and suitable for *Atmospheric Chemistry and Physics*. The authors' main finding is that OH oxidation primarily leads to bisulfate ion formation, and propose an H-atom abstraction pathway by the sulfate radical anion. The chemistry is plausible, but as described in my first comment, the authors could have discussed other mechanisms. In general, I think the manuscript is well written and should be published after addressing my comments.

**Major Comments**

1.  The authors propose that bisulfate ion formation is primarily due to sulfate ion abstraction of hydrogen. First, it would help if the authors included a reaction mechanism in addition to the mechanisms shown in Scheme 1 and Scheme 2, showing the different products formed after hydrogen abstraction by the sulfate ion. However, in analogy to the OH-initiated oxidation of simple alkyl esters (see e.g., Sun et al., 2012), instead of H-atom abstraction by sulfate ion, might there be rearrangement of the hydrogen atom from the alkoxy radical carbon to the oxygen on the methoxy group, and decomposition of the O-C bond from the methoxy group to make formyl radical (HCO)? Subsequent reactions involving HCO would form CO and could explain the absence of additional products, besides $HSO_4^-$, after oxidation of sodium methyl sulfate.

2.  These experiments were conducted in the absence of $NO_x$, which may be more relevant for pristine, low $NO_x$ environments. Could the authors place this chemistry in context with varying ratios of $NO_x$ and $RO_2$? How might the schemes differ under high/low NO?

**Minor Comments**

1.  The manuscript discussion seems equally focused on sodium *methyl* sulfate and sodium *ethyl* sulfate. The authors might think of including both in the title.

2.  Section 2 (lines 17-19): I am not convinced that the following statement is true, "The sodium methyl sulfate has a low estimated vapor pressure of $4.65 \times 10^{-2}$ mmHg (Chemistry Dashboard), and therefore, volatilization and gas-phase oxidation of sodium

methyl sulfate are expected to be insignificant in these experiments." Such a vapor pressure is ~60 ppm. The authors should clarify that if gas phase oxidation takes place, what effect it could have on the results?

3. What potential effects are there, if any, from the exposure of sodium methyl sulfate to ozone and UV light from the $O_3$ lamp?

**References**

Sun, Y. et al., "Mechanism and kinetic studies for OH radical-initiated atmospheric oxidation of methyl propionate", *Atmos. Environ.*, **63**, 14-21, 2012.

---

## Referee Comment (RC2) · Anonymous Referee #1 · 21 Nov 2017

This is an interesting manuscript describing the formation of sulfate radical anions through heterogeneous oxidation and transformation of selected organosulfate model compounds (sodium methyl sulfate and sodium ethyl sulfate). The authors reported kinetics and mass spectrometric data to explain the potential chemical mechanisms for the observed composition change. Overall, this manuscript is concise and well written, and provides useful information for a better understanding of the chemical evolution of particle phase products that is thus far not very well understood. These findings are of great interest to the atmospheric chemistry community. I am in support of publication

after the following comments being addressed in the revised manuscript.

1) Have the authors observed any evidence for hydrolysis of organosulfates without OH exposures? Could the presence of bisulfate be also resulting from the acid-base equilibrium of sulfate ions in the aqueous droplet (via water dissociation) without radicals involved? 2) Page 3, Line 17: the estimated vapor pressure for sodium methyl sulfate is actually not low enough that can be considered as insignificant. Please provide more support to justify this statement. In addition, is there any evidence showing that this compound may be present in the gas phase? 3) Page 6, Lines 13-14: what are the bond dissociation energies for the $C-O$ versus $C-C$ bonds, respectively? 4) Page 6, Lines 14-15: please provide more detailed explanation about why the authors think the bond dissociation energy of a $C-O$ bond is likely to be lowered in the presence of a sulfur atom ($C-O-S$). 5) Page 6, Line 23: "......typically have an order of magnitude ranging from......" This sentence reads incomplete. Please double check if something is missing. 6) Page 6, Line 27: the acid dissociation constant (Ka) of the bisulfate ion is $1.2 \times 10^{-2}$. The authors might report this as "pKa" by mistake.

---

## Referee Comment (RC3) · Anonymous Referee #2 · 19 Dec 2017

Kwong et al. present studies detailing the heterogenous OH oxidation of sodium methyl sulfate ($CH_3SO_4Na$) and sodium ethyl sulfate ($C_2H_5SO_4Na$) particles. The particles were exposed to OH radicals in a well-characterized flow reactor that has been applied in many heterogenous oxidation kinetic studies over the last 10 years or so. Ensemble aerosol mass spectra were obtained with DART. The organic sulfates were found to have effective rate constants ranging from $(3.79-4.64)*10^{-13}$ $cm^3$ $molec^{-1}$ $s^{-1}$ and effective uptake coefficients ranging from 0.17-0.19. For both organic sulfates, $HSO_4^-$ was detected as an oxidation product, implying the formation of aqueous sulfate radical. Proposed reaction schemes detailing functionalization and fragmentation-dominated reaction pathways are examined. Overall, the experiments seemed to have been done carefully and the results are presented concisely. However, in my opinion more effort needs to be made to place these results in atmospheric context before I could support publication in ACP.

**Main comments**

1. An effective OH rate constant of $\sim4*10^{-13}$ $cm^3$ $molec^{-1}$ $s^{-1}$ corresponds to an OH oxidation lifetime of nearly 20 days assuming a 24-hour average OH mixing ratio of $1.5*10^6$ $cm^{-3}$. As far as I can tell, this timescale is too long to compete with other removal processes for organic sulfates such as wet or dry deposition. If anything, the results suggest to me that, to a first approximation, it is reasonable to treat organic sulfates as chemically stable over atmospherically relevant timescales. However, If the authors are aware of atmospheric measurements that suggest atmospheric degradation of organic sulfates, that would be a strong addition to the paper and further motivate the present work.

2. The authors propose that sulfate radicals generated from OH oxidation of organic sulfates could contribute to secondary condensed-phase chemistry, but the relevant oxidation timescales are not discussed. They state that sulfate rate constants range from $10^6$ - $10^9$ $M^{-1}$ $s^{-1}$, and imply that because these are comparable to OH rate constants ($10^7$ to $10^{11}$ $M^{-1}$ $s^{-1}$), secondary sulfate chemistry might be important. I disagree with this conclusion because this cited range of sulfate rate constants is one to two orders' magnitude lower than OH rate constants. Furthermore, it is not appropriate to make this conclusion without citing corresponding ranges of condensed phase sulfate vs OH radical concentrations, and calculating the corresponding oxidation lifetimes with respect to sulfate radicals vs OH radicals. If the authors are aware of atmospheric measurements that suggest influence of sulfate radical-initiated oxidation chemistry, that would be a strong addition to the paper and further motivate the present work

**Additional/Minor comments**

3. I did not notice discussion of control experiments detailing exposure to ozone or 254 nm photons to rule out the contributions of ozonolysis or photolysis to the degradation of organic sulfates.

4. A short description of the DART technique should be added to the experimental section for readers that are not familiar with the method (including this reviewer).

5. **P3, L17**: I calculate that a vapor pressure of 0.0465 torr corresponds to a saturation concentration of 3.3E5 ug/m3 at 20 deg C and 1 atm. If that vapor pressure is accurate, this statement is likely not true: "volatilization and gas-phase oxidation of sodium methyl sulfate are expected to be insignificant in these experiments." (The range of mass concentration is not provided, but I am assuming it is far less than $3*10^5$ ug/m3, although this should be clarified). Some other more plausible evidence is needed to support the claim that volatilization/gas-phase oxidation is unimportant.

6. **P3, L30**: Please clarify how the was the residence time of 1.3 min was determined, i.e., calculated from reactor volume and flow rate or obtained from a measured residence time distribution.

7. **P5, L13**: typo ("occur" → "occurring")

8. **P6, L23**: change "have an order of magnitude ranging" to "values"

9. Figures 1 and 5, 2 and 8, 3 and 6, and 4 and 9, and corresponding discussion, could easily be merged/consolidated to decrease the number of figures.

---

## Author Comment (AC2) · 17 Jan 2018

*In this paper, titled "Importance of sulfate radical anion formation and chemistry in heterogeneous OH oxidation of sodium methyl sulfate, the smallest organosulfate" by Kwong et al., the authors present an interesting dataset focused on the heterogeneous chemical transformation of an organic sulfate compound. Organosulfates often have low saturation vapor pressure and have been identified in ambient aerosol particles, and therefore are considered important for SOA formation in the atmosphere. However, there is currently very little knowledge of the particle phase transformations of organosulfates, which could be important because variations in the organosulfate composition may influence particle volatility and SOA mass. Therefore, the topic is very much atmospherically relevant and suitable for Atmospheric Chemistry and Physics. The authors' main finding is that OH oxidation primarily leads to bisulfate ion formation, and propose an H-atom abstraction pathway by the sulfate radical anion. The chemistry is plausible, but as described in my first comment, the authors could have discussed other mechanisms. In general, I think the manuscript is well written and should be published after addressing my comments.*

**We are grateful for the comments from the reviewer and sincerely thank him/her for suggestions. Please see our responses to the comments and suggestions below.**

Major Comments:
**Reviewer Comments #1**
*The authors propose that bisulfate ion formation is primarily due to sulfate ion abstraction of hydrogen. First, it would help if the authors included a reaction mechanism in addition to the mechanisms shown in Scheme 1 and Scheme 2, showing the different products formed after hydrogen abstraction by the sulfate ion. However, in analogy to the OH-initiated oxidation of simple alkyl esters (see e.g., Sun et al., 2012), instead of H-atom abstraction by sulfate ion, might there be rearrangement of the hydrogen atom from the alkoxy radical carbon to the oxygen on the methoxy group, and decomposition of the O−C bond from the methoxy group to make formyl radical (HCO)? Subsequent reactions involving HCO would form CO and could explain the absence of additional products, besides $HSO_4^-$, after oxidation of sodium methyl sulfate.*

**Author Response**

We agree with the reviewer's suggestion on the reaction mechanisms for the OH oxidation of sodium methyl sulfate and the formation of bisulfate ion. This proposed mechanism is discussed in the main text and presented in the **Scheme 1** in the revised manuscript.

Page 6, Lines 18-21

"Alternatively, the decomposition of the alkoxy radical, in analogy to the OH radical-initiated oxidation of simple alkyl esters (*Sun et al., 2012*), could involve the rearrangement of the hydrogen atom from the alkoxy radical carbon to the oxygen on the methoxy group. The decomposition of the C−O bond from the methoxy group generates a bisulfate ion and a formyl radical (CHO). The subsequent reactions of the formyl radical can yield a carbon monoxide (CO) and $HO_2$. Like formaldehyde, carbon monoxide is volatile and partitions back to the gas phase."

**Reviewer Comments #2**

*These experiments were conducted in the absence of $NO_x$, which may be more relevant for pristine, low NOx environments. Could the authors place this chemistry in context with varying ratios of $NO_x$ and $RO_2$? How might the schemes differ under high/low NO?*

**Author Response**

We postulate that under high NOx environment (i.e. high NO), $RO_2$ + NO reactions might become more favorable compared to $RO_2$ + $RO_2$ reactions. Alkoxy radicals are more likely formed from the $RO_2$ + NO reactions and subsequently decompose to yield the volatile products (formaldehyde and carbon monoxide), sulfate radicals, and bisulfate ions as proposed in **Scheme 1**. Generally, the increase in NOx concentration likely favors the formation and decomposition of alkoxy radicals during the oxidation.

**Minor Comments:**

**Reviewer Comments #3**

*The manuscript discussion seems equally focused on sodium methyl sulfate and sodium ethyl sulfate. The authors might think of including both in the title.*

**Author Response**

We would like to thank the reviewer's suggestion. We decide to keep the title unchanged because we

would like to focus the discussion on the heterogeneous OH oxidation of sodium methyl sulfate, which has been detected in atmospheric aerosols (*Hettiyadura et al., 2015*).

**Reviewer Comments #4**

*Section 2 (lines 17-19): I am not convinced that the following statement is true, "The sodium methyl sulfate has a low estimated vapor pressure of 4.65 × 10⁻² mmHg (Chemistry Dashboard), and therefore, volatilization and gas-phase oxidation of sodium methyl sulfate are expected to be insignificant in these experiments." Such a vapor pressure is ~ 60 ppm. The authors should clarify that if gas phase oxidation takes place, what effect it could have on the results?*

**Author Response**

We agree with reviewer's comment and would like to note that the other two reviewers have raised the same comment on the potential volatilization and gas phase oxidation of sodium methyl sulfate (reviewer 1, comments #2 and reviewer 2, comments # 5). To be concise and address the volatility issue clearly once in the response, please kindly refer to our response to reviewer 1 (reviewer 1, comments #2) on this topic.

**Reviewer Comments #5**

*What potential effects are there, if any, from the exposure of sodium methyl sulfate to ozone and UV light from the O3 lamp?*

**Author Response**

We have run control experiments to investigate the potential volatilization of parent compound (i.e. sodium methyl sulfate) and the effects of ozone and UV light on the composition of the aerosols before oxidation under the same experimental conditions. We found that there is no change in aerosol mass spectra in the presence of ozone without the UV light, suggesting that the reaction of sodium methyl sulfate with ozone is not significant. The aerosol mass spectrum is about the same as that obtained in the absence of ozone with the UV light, suggesting that the photolysis of sodium methyl sulfate aerosols is not likely. We have added this information in the revised manuscript.

Page 4, Lines 18-26

"We have run control experiments to investigate the potential volatilization of parent compound (i.e. sodium methyl sulfate), and the effects of ozone and UV light on the composition of the aerosols before oxidation under the same experimental conditions. To investigate the volatilization of sodium methyl sulfate, we have measured the mass spectrum by filtering out the aerosols, and the parent peak is very small, suggesting there is very small amount of sodium methyl sulfate present in the gas phase. Volatilization and gas-phase oxidation of sodium methyl sulfate is expected to be not significant. For the effect of ozone and UV light, we found that there is no change in aerosol mass

spectra in the presence of ozone without the UV light, suggesting that the reaction of sodium methyl sulfate with ozone is not significant. The aerosol mass spectrum is about the same as that obtained in the absence of ozone with the UV light, suggesting that the photolysis of sodium methyl sulfate aerosols is not likely to occur."

**References**

Hettiyadura, A. P. S., Stone, E. A., Kundu, S., Baker, Z., Geddes, E., Richards, K., Humphry, T.: Determination of Atmospheric Organosulfates using HILIC Chromatography with MS Detection, *Atmos. Meas. Tech.*, 8(6), 2347−2358, **2015**.

Sun, X., Hu, Y., Xu, F., Zhang, Q., Wang, W.: Mechanism and Kinetic Studies for OH Radical-initiated Atmospheric Oxidation of Methyl Propionate, *Atmos. Environ.*, 63, 14−21, **2012**.

[Figure]

**Scheme 1. Proposed reaction mechanism for heterogeneous OH oxidation of sodium methyl sulfate (Gray arrows denote the minor pathways)**

[Figure]

**Scheme 2. Proposed reaction mechanism of heterogeneous OH oxidation of sodium ethyl sulfate (Gray arrows denote the minor pathways)**

---

## Author Comment (AC3) · 17 Jan 2018

*This is an interesting manuscript describing the formation of sulfate radical anions through heterogeneous oxidation and transformation of selected organosulfate model compounds (sodium methyl sulfate and sodium ethyl sulfate). The authors reported kinetics and mass spectrometric data to explain the potential chemical mechanisms for the observed composition change. Overall, this manuscript is concise and well written, and provides useful information for a better understanding of the chemical evolution of particle phase products that is thus far not very well understood. These findings are of great interest to the atmospheric chemistry community. I am in support of publication after the following comments being addressed in the revised manuscript.*

**We are grateful for the comments from the reviewer and sincerely thank him/her for the suggestions. Please see our responses to the comments and suggestions below.**

**General Comments:**
**Reviewer Comments #1**
*Have the authors observed any evidence for hydrolysis of organosulfates without OH exposures? Could the presence of bisulfate be also resulting from the acid-base equilibrium of sulfate ions in the aqueous droplet (via water dissociation) without radicals involved?*

**Author Response**
In the literature, Hu et al. (2011) have investigated the thermodynamics and kinetics of the hydrolysis of atmospherically relevant organonitrates and organosulfates. They found that primary and secondary organosulfates are stable against hydrolysis under atmospherically relevant aerosol acidities and lifetimes, while tertiary organosulfates may undergo hydrolysis efficiently. Since the sodium methyl sulfate and sodium ethyl sulfate are primary organosulfates, we expect the hydrolysis would not be significant in our experiments and bisulfate is not likely originated from the acid-base equilibrium of sulfate ions in the aqueous droplet (via water dissociation).

**Reviewer Comments #2**
*Page 3, Line 17: the estimated vapor pressure for sodium methyl sulfate is actually not low enough that can be considered as insignificant. Please provide more support to justify this statement. In*

*addition, is there any evidence showing that this compound may be present in the gas phase?*

**Author Response**

We agree with reviewer's comment. To our best knowledge, the vapor pressure of sodium methyl sulfate has not been experimentally measured. We would like to acknowledge that the value reported in the original manuscript is estimated using a model developed by the United States Environmental Protection Agency. In addition to heterogeneous OH oxidation experiments, we have run control experiments to investigate the potential volatilization of parent compound (i.e. sodium methyl sulfate), and the effects of ozone and UV light on the sodium methyl sulfate before oxidation under the same experimental conditions. To investigate the volatilization of sodium methyl sulfate, we have measured the mass spectrum by filtering out the aerosols. As shown in the mass spectrum below (**Figure S1**), the parent peak is very small and is about 20 times less than that obtained before oxidation (**Figure 1**).

[Figure]

**Figure S1. Mass spectrum without sodium methyl sulfate aerosols.**

This suggests that there is insignificant amount of sodium methyl sulfate present in the gas phase. We would also like to mention that organosulfates and their salts are in general considered to have low volatilities and have not been detected in the gas phase in field studies. These results suggest that volatilization and gas phase oxidation of sodium methyl sulfate is not significant. To avoid confusion, we have removed the estimated vapor pressure of sodium methyl sulfate. We have added the following information in the revised manuscript to show that volatilization and gas phase oxidation of sodium methyl sulfate is likely not significant in this study.

Page 4, Lines 18-22

"We have run control experiments to investigate the potential volatilization of parent compound (i.e. sodium methyl sulfate), and the effect of ozone and UV light on the composition of the aerosols before oxidation under the same experimental conditions. To investigate the volatilization of sodium methyl sulfate, we have measured the mass spectrum by filtering out the aerosols, and the parent peak is very small, suggesting there is very small amount of sodium methyl sulfate present in the gas phase. Volatilization and gas-phase oxidation of sodium methyl sulfate is expected to be not significant."

**Reviewer Comments #3**

*Page 6, Lines 13-14: what are the bond dissociation energies for the C−O versus C−C bonds, respectively?*

**Author Response**

We would like to acknowledge that the dissociation energies for C−O and C−C bonds of sodium methyl sulfate are not known. However, the dissociation energy of C−O bond is in general thought to be smaller than that of C−C bond (*Dean and Lang, 1992*). We have clarified this point in the revised manuscript.

Page 7, Lines 7-10

"Although the dissociation energies for C−O and C−C bonds of sodium methyl sulfate are not known, fragmentation processes could be enhanced since the decomposition of the alkoxy radical involves the cleavage of a C−O bond, which is in general thought to be weaker than a C−C bond (*Dean and Lang, 1992*)."

**Reviewer Comments #4**

*Page 6, Lines 14-15: please provide more detailed explanation about why the authors think the bond dissociation energy of a C−O bond is likely to be lowered in the presence of a sulfur atom (C−O−S).*

**Author Response**

Oae and Doi (1991) and Dean and Lange (1992) have reported that dissociation energies for different kinds of bonds and show the following trend: C−C > C−OC > C−SC > C−OS. The bond dissociation energy of a C−O bond is smaller in the presence of a sulfur atom or sulfur-oxygen bearing group compared to a carbon atom (C−OC > C−OS). One possible explanation is that sulfur atom or sulfur-oxygen bearing group (e.g. sulfate) is more electronegative than a carbon atom, reducing the electron density and bond strength of the C−O bond by inductive effect. We acknowledge that further investigation is required to better understand the effect of sulfate group on the dissociation energies of C−O bond for the sodium methyl sulfate. We have revised the sentences in the revised

manuscript to reflect the unknown effect of sulfate group on the decomposition of the C−O bond in the sodium methyl sulfate.

Page 7, Lines 10-14

"The bond dissociation energy of a C−O bond might be lowered in the presence of a sulfur atom or sulfur-oxygen bearing group (*Oae and Doi, 1991; Dean and Lange, 1992*). One possibility is that sulfur atom or sulfur-oxygen bearing group (e.g. sulfate) is more electronegative than a carbon atom, reducing the electron density and bond strength of the C−O bond by inductive effect. Further investigation is required to better understand the effect of sulfate group on the dissociation energies of the C−O bond for the sodium methyl sulfate."

**Reviewer Comments #5**

*Page 6, Line 23: "…… typically have an order of magnitude ranging from ……" This sentence reads incomplete. Please double check if something is missing.*

**Author Response**

We have revised the sentence.

Page 7, Lines 20-22

"The second-order rate constants for $SO_4^{\bullet-}$ reactions with alcohols, ethers, alkanes, and aromatic compounds typically range in value from $10^6$ to $10^9$ $M^{-1}$ $s^{-1}$ (*Clifton and Huie, 1989; Neta et al., 1977; Neta et al., 1988; Padmaja et al., 1993*)."

**Reviewer Comments #6**

*Page 6, Line 27: the acid dissociation constant ($K_a$) of the bisulfate ion is $1.2 \times 10^{-2}$. The authors might report this as "$pK_a$" by mistake.*

**Author Response**

Thank you for pointing out the typo. We have corrected it in the revised manuscript.

Page 7, Lines 26-29

"For the OH reaction with sodium methyl sulfate, it is proposed that sulfate radical anion, once formed, can abstract a hydrogen atom from the neighboring, unreacted sodium methyl sulfate, yielding the bisulfate ion, which has a small acid dissociation constant in equilibrium with sulfate ($SO_4^{2-}$) and hydrogen ($H^+$) ions ($K_a = 1.2 \times 10^{-2}$) (*Brown et al., 2012*)."

**References**

Dean, J. and Lange, N.: Lange's handbook of chemistry (15[th] Edition). *New York: McGraw-Hill,*

4.42−4.43, **1992**.

Hu, K. S., Darer, A. I., Elrod, M. J.: Thermodynamics and kinetics of the hydrolysis of atmospherically relevant organonitrates and organosulfates, *Atmos. Chem. Phys.*, 11, 8307−8320, **2011**.

Oae, S. and Doi, J.: Organic Sulfur Chemistry: Structure and Mechanism, *CRC Press: Boca Raton, Florida*, **1991**.

---

## Author Comment (AC4) · 17 Jan 2018

*Kwong et al. present studies detailing the heterogeneous OH oxidation of sodium methyl sulfate (CH$_3$SO$_4$Na) and sodium ethyl sulfate (C$_2$H$_5$SO$_4$Na) particles. The particles were exposed to OH radicals in a well-characterized flow reactor that has been applied in many heterogeneous oxidation kinetic studies over the last 10 years or so. Ensemble aerosol mass spectra were obtained with DART. The organic sulfates were found to have effective rate constants ranging from $(3.79 - 4.64) \times 10^{-13}$ cm$^3$ molecule$^{-1}$ s$^{-1}$ and effective uptake coefficients ranging from $0.17 - 0.19$. For both organic sulfates, HSO$_4^-$ was detected as an oxidation product, implying the formation of aqueous sulfate radical. Proposed reaction schemes detailing functionalization and fragmentation-dominated reaction pathways are examined. Overall, the experiments seemed to have been done carefully and the results are presented concisely. However, in my opinion more effort needs to be made to place these results in atmospheric context before I could support publication in ACP.*

**We are grateful for the comments from the reviewer and sincerely thank him/her for suggestions. Please see our responses to the comments and suggestions below.**

Major Comments:
**Reviewer Comments #1**
*An effective OH rate constant of ~ 4\*10$^{-13}$ cm$^3$ molec$^{-1}$ s$^{-1}$ corresponds to an OH oxidation lifetime of nearly 20 days assuming a 24-hour average OH mixing ratio of 1.5\*10$^6$ cm$^{-3}$. As far as I can tell, this timescale is too long to compete with other removal processes for organic sulfates such as wet or dry deposition. If anything, the results suggest to me that, to a first approximation, it is reasonable to treat organic sulfates as chemically stable over atmospherically relevant timescales. However, If the authors are aware of atmospheric measurements that suggest atmospheric degradation of organic sulfates, that would be a strong addition to the paper and further motivate the present work.*

**Author Response**
We agree with the reviewer's comment. Assuming a 24-hour average OH mixing ratio of $1.5 \times 10^6$ cm$^{-3}$, the lifetime of sodium methyl sulfate against heterogeneous OH oxidation is about 20 days. This timescale is longer than other removal processes such as wet or dry deposition. To our best knowledge, laboratory studies have revealed that primary and secondary organosulfates are stable

against hydrolysis under atmospheric-relevant aerosol acidities and lifetimes, while tertiary organosulfates may undergo hydrolysis efficiently (*Hu et al., 2011*). Since sodium methyl sulfate is a primary organosulfates, it is expected to be stable against hydrolysis. These results suggest that sodium methyl sulfate is likely chemically stable over atmospherically relevant timescales. We would also like to note that studying the heterogeneous reactivity of sodium methyl sulfate towards OH radicals provides a much needed fundamental understanding of the oxidation kinetics and pathways, and these data may be applied in the interpretation of the oxidation of more complex organosulfates, which may have a range of chemical lifetimes in the atmosphere. This information is added in the revised manuscript.

Page 5, Lines 13-21

"Assuming a 24-hour average OH concentration of $1.5 \times 10^6$ cm$^{-3}$, the lifetime of sodium methyl sulfate against heterogeneous OH oxidation is about 20 days. This timescale is longer than other removal processes such as wet or dry deposition. Laboratory studies have revealed that primary and secondary organosulfates are stable against hydrolysis under atmospheric relevant aerosol acidities and lifetimes, while tertiary organosulfates may undergo hydrolysis efficiently (*Hu et al., 2011*). Since sodium methyl sulfate is a primary organosulfate, it is expected to be stable against hydrolysis. These results suggest that sodium methyl sulfate is likely chemically stable over atmospherically relevant timescales. Studying the heterogeneous reactivity of sodium methyl sulfate towards OH radicals provides a much needed fundamental understanding of the oxidation kinetics and pathways, and these data may be applied in the interpretation of the oxidation of more complex organosulfates, which may have a range of chemical lifetimes in the atmosphere."

**Reviewer Comments #2**

*The authors propose that sulfate radicals generated from OH oxidation of organic sulfates could contribute to secondary condensed-phase chemistry, but the relevant oxidation timescales are not discussed. They state that sulfate rate constants range from $10^6 - 10^9$ M$^{-1}$ s$^{-1}$, and imply that because these are comparable to OH rate constants ($10^7$ to $10^{11}$ M$^{-1}$ s$^{-1}$), secondary sulfate chemistry might be important. I disagree with this conclusion because this cited range of sulfate rate constants is one to two orders' magnitude lower than OH rate constants. Furthermore, it is not appropriate to make this conclusion without citing corresponding ranges of condensed phase sulfate vs OH radical concentrations, and calculating the corresponding oxidation lifetimes with respect to sulfate radicals vs OH radicals. If the authors are aware of atmospheric measurements that suggest influence of sulfate radical-initiated oxidation chemistry, that would be a strong addition to the paper and further motivate the present work.*

**Author Response**

We would like to thank the reviewer's comment and agree that the reported SO$_4$$^{\bullet-}$ rate constants and

OH rate constants are not comparable. In the literature, the typical aqueous phase radical concentration of OH and $SO_4^{\bullet-}$ are of $10^{-13}$ M and $10^{-14}$ M, respectively (*Lelieveld and Crutzen, 1991; Herrmann et al., 2000*). With the reported OH rate constants ($10^7 - 10^{11}$ $M^{-1}$ $s^{-1}$) and $SO_4^{\bullet-}$ rate constants ($10^6 - 10^9$ $M^{-1}$ $s^{-1}$), the calculated lifetime toward aqueous phase oxidation with OH and $SO_4^{\bullet-}$ is about 2 minutes − 12 days and 1.2 days − 3 years, respectively. These results suggest that aqueous phase OH radical reactions with organic compounds are likely the dominant reaction pathways. Some organic compounds (e.g. alkanes and alkenes) are stable against $SO_4^{\bullet-}$ initiated oxidation, but some (e.g. alcohols and ethers) react with $SO_4^{\bullet-}$ at atmospherically relevant timescale. We would like to note that $SO_4^{\bullet-}$ reaction rate with sodium methyl sulfate is not known. Future works are needed to better understand the role of the $SO_4^{\bullet-}$ initiated oxidation chemistry in chemical transformation of sodium methyl sulfate and organic compounds in the atmospheric aerosols. We have clarified this point in the revised manuscript.

Page 7, Lines 20-26

"The second-order rate constants for $SO_4^{\bullet-}$ reactions with alcohols, ethers, alkanes, and aromatic compounds typically range in value from $10^6$ to $10^9$ $M^{-1}$ $s^{-1}$ (*Clifton and Huie, 1989; Neta et al., 1977; Neta et al., 1988; Padmaja et al., 1993*). With an aqueous phase $SO_4^{\bullet-}$ concentration of $10^{-14}$ M (*Herrmann et al., 2000*), the calculated lifetime toward aqueous phase oxidation with $SO_4^{\bullet-}$ ranges from 1.2 days to 3 years. These results suggest that some organic compounds (e.g. alkanes and alkenes) are stable against $SO_4^{\bullet-}$ initiated reactions, but some (e.g. alcohols and ethers) can react with $SO_4^{\bullet-}$ efficiently. Future works are needed to better understand the role of $SO_4^{\bullet-}$ initiated oxidation chemistry in chemical transformation of sodium methyl sulfate and organic compounds in the atmospheric aerosols."

**Minor Comments:**
**Reviewer Comments #3**
*I did not notice discussion of control experiments detailing exposure to ozone or 254 nm photons to rule out the contributions of ozonolysis or photolysis to the degradation of organic sulfates.*

**Author Response**
We have run control experiments to investigate the potential volatilization of parent compound (i.e. sodium methyl sulfate) and the effects of ozone and UV light on the composition of the aerosols before oxidation under the same experimental conditions. We found that there is no change in aerosol mass spectra in the presence of ozone without the UV light, suggesting that the reaction of sodium methyl sulfate with ozone is not significant. The aerosol mass spectrum is about the same as that obtained in the absence of ozone with the UV light, suggesting that the photolysis of sodium methyl sulfate aerosols is not likely. We have added this information in the revised manuscript.

Page 4, Lines 18-26

"We have run control experiments to investigate the potential volatilization of parent compound (i.e. sodium methyl sulfate), and the effects of ozone and UV light on the composition of the aerosols before oxidation under the same experimental conditions. To investigate the volatilization of sodium methyl sulfate, we have measured the mass spectrum by filtering out the aerosols, and the parent peak is very small, suggesting there is very small amount of sodium methyl sulfate present in the gas phase. Volatilization and gas-phase oxidation of sodium methyl sulfate is expected to be not significant. For the effect of ozone and UV light, we found that there is no change in aerosol mass spectra in the presence of ozone without the UV light, suggesting that the reaction of sodium methyl sulfate with ozone is not significant. The aerosol mass spectrum is about the same as that obtained in the absence of ozone with the UV light, suggesting that the photolysis of sodium methyl sulfate aerosols is not likely to occur."

**Reviewer Comments #4**

*A short description of the DART technique should be added to the experimental section for readers that are not familiar with the method (including this reviewer).*

**Author Response**

We have added a brief description of DART in the revised manuscript.

Page 4, Lines 8-16

"The details of the DART operation have been described elsewhere (*Cody et al., 2005*). The DART ionization source was operated in the negative-ion mode. Helium was chosen as the ionizing gas and entered an ionization chamber, where a high electric potential of 4 kV was applied. This generates a glow discharge containing ions, electrons, and metastable helium atoms. A potential of 200 V was applied to two electrostatic lenses to remove ions and only the metastable helium atoms exited the chamber. The gas stream was heated to 500 ℃ before leaving the ionization source. The metastable helium atoms are responsible for ionizing the gas-phase species in the ionization region (*Chan et al., 2014; Cheng et al., 2015, 2016*). For ionic compounds like sodium methyl sulfate, negative ions can be formed via direct ionization in the negative ion mode (*Hajslova et al., 2011*); for instance, pyruvate ions have been detected from ammonium pyruvate using the DART (*Block et al., 2010*)."

**Reviewer Comments #5**

*P3, L17: I calculate that a vapor pressure of 0.0465 torr corresponds to a saturation concentration of $3.3E5$ $ug/m^3$ at 20 deg C and 1 atm. If that vapor pressure is accurate, this statement is likely not true: "volatilization and gas-phase oxidation of sodium methyl sulfate are expected to be insignificant in these experiments." (The range of mass concentration is not provided, but I am assuming it is far less than $3*10^5$ ug/m3, although this should be clarified). Some other more*

*plausible evidence is needed to support the claim that volatilization/gas-phase oxidation is unimportant.*

**Author Response**

We agree with reviewer's comment and would like to note that the other two reviewers have raised the same comment on the potential volatilization and gas phase oxidation of sodium methyl sulfate (reviewer 1, comments #2 and reviewer 3, comments # 4). To be concise and address the volatility issue clearly once in the response, please kindly refer to our response to reviewer 1 (reviewer 1, comments #2) on this topic.

**Reviewer Comments #6**

*P3, L30: Please clarify how the was the residence time of 1.3 min was determined, i.e., calculated from reactor volume and flow rate or obtained from a measured residence time distribution.*

**Author Response**

We obtained this value from measured aerosol residence time distribution.

**Reviewer Comments #7**

*P5, L13: typo ("occur" → "occurring")*

**Author Response**

Thank you for pointing out the typo. We have revised the sentence.

Page 6, Lines 4-6

"Using **Eqn. 3**, the $\gamma_{eff}$ is calculated to be 0.17 ± 0.03. Although the $\gamma_{eff}$ is less than 1, as will be discussed in the section 3.3, secondary reactions are likely occurring, leading to the formation and subsequent reactions of sulfate radical anions ($SO_4^{\bullet-}$)."

**Reviewer Comments #8**

*P6, L23: change "have an order of magnitude ranging" to "values"*

**Author Response**

Thank you for the suggestion. We have revised the sentence.

Page 7, Lines 20-22

"The second-order rate constants for $SO_4^{\bullet-}$ reactions with alcohol, ethers, alkanes, and aromatic compounds typically range in value from $10^6$ to $10^9$ $M^{-1}$ $s^{-1}$ (*Clifton and Huie, 1989; Neta et al., 1977; Neta et al., 1988; Padmaja et al., 1993*)."

**Reviewer Comments #9**

*Figures 1 and 5, 2 and 8, 3 and 6, and 4 and 9, and corresponding discussion, could easily be merged/consolidated to decrease the number of figures.*

**Author Response**

We would like to thank the reviewer's suggestion but would like to keep the figures and corresponding discussion separately for the following reasons. First, we would like to focus the discussion on sodium methyl sulfate, which is the smallest organosulfate detected in atmospheric aerosols (as sodium ethyl sulfate has not been found in the atmosphere). Second, the results of sodium ethyl sulfate are primarily used to confirm the detectability of functionalization products using the DART ionization source coupled with a high-resolution mass spectrometer. Third, as the ionization efficiency and the relative abundance of two alkyl sulfates and their reaction products are not known, we would like to present the results of these two organosulfates in separate figures.

**References**

Block, E., Dane, A. J., Thomas, S., Cody, R. B.: Applications of Direct Analysis in Real Time Mass Spectrometry (DART-MS) in Allium Chemistry. 2-Propenesulfenic and 2-Propenesulfinic Acids, Diallyl Trisulfane S-Oxide, and Other Reactive Sulfur Compounds from Crushed Garlic and Other Alliums, *J. Agric. Food Chem.*, 58(8), 4617−4625, **2010**.

Chan, M. N., Zhang, H., Goldstein, A. H., Wilson, K. R.: The Role of Water and Phase in the Heterogeneous Oxidation of Solid and Aqueous Succinic Acid Aerosol by Hydroxyl Radicals, *J. Phys. Chem. C*, 118(50), 28978–28992, **2014**.

Cheng, C. T., Chan, M. N., Wilson, K. R.: The Role of Alkoxy Radicals in the Heterogeneous Reaction of Two Structural Isomers of Dimethylsuccinic Acid, *Phys. Chem. Chem. Phys.*, 17(38), 25309–25321, **2015**.

Cheng, C. T., Chan, M. N., Wilson, K. R.: Importance of Unimolecular $HO_2$ Elimination in the Heterogeneous OH Reaction of Highly Oxygenated Tartaric Acid Aerosol, *J. Phys. Chem. A*, 120(29), 5887–5896, **2016**.

Cody, R. B., Laramee, J. A., Durst, H. D.: Versatile New Ion Source for the Analysis of Materials in Open Air under Ambient Conditions, *Anal. Chem.*, 77 (8), 2297−2302, **2005**.

Hajslova, J., Cajka, T., Vaclavik, L.: Challenging Applications Offered by Direct Analysis in Real Time (DART) in Food-Quality and Safety Analysis, Trends Anal. Chem., 30(2), 204−218, 2011.
Herrmann, H., Ervens, B., Jacobi, H. W., Wolke, R., Nowacki, P., Zellner, R.: CAPRAM2.3: A

Chemical Aqueous Phase Radical Mechanism for Tropospheric Chemistry, *J. Atmos. Chem.*, 36, 231−284, **2000**.

Hu, K. S., Darer, A. I., Elrod, M. J.: Thermodynamics and Kinetics of the Hydrolysis of Atmospherically Relevant Organonitrates and Organosulfates, *Atmos. Chem. Phys.*, 11, 8307−8320, **2011**.

Lelieveld, J., Crutzen, P. J.: The Role of Clouds in Tropospheric Photochemistry, *J. Atmos. Chem.*, 12, 229−267, **1991**.